## [Transparent Peer Review file · Nature Communications]

Hh and EGFR-Ras signaling promote distinct steps of tumor progression in the *Drosophila* follicle epithelium

Corresponding Author: Dr Katja Rust

Version 0:

Reviewer comments:

Reviewer #1

(Remarks to the Author)

Reviewer Comments on "A single-cell analysis of Hh and EGFR-Ras: Independent signaling pathways prevent tumorigenesis"

Rust et al. present a technically sophisticated study combining genetic perturbation and single-cell transcriptomics to explore how Hh and EGFR-Ras signaling affect cell fate and proliferation in the *Drosophila* follicle epithelium. The authors identify transcriptional hybrid states, altered cell cycle control, and overgrowth phenotypes under pathway co-activation, which they interpret as hallmarks of tumorigenesis. While the dataset is rich and the experimental system well-chosen, the study's interpretive clarity and mechanistic depth are hampered by key conceptual, analytical, and structural weaknesses.

Major Points

1. The manuscript suffers from a lack of a clear conceptual thread. The experimental flow often feels observational rather than hypothesis-driven, with new datasets introduced without clear rationale. This impairs readability and interpretability, even for domain experts. To improve clarity and engagement, the authors should reorganize the results around discrete, testable questions and explicitly state the purpose of each experimental step.
2. The assertion that Hh and EGFR-Ras function independently is not supported by mechanistic data. The pathways converge on shared transcriptional targets and exhibit synergistic phenotypes. Without epistasis experiments or systematic pathway activity measurements (e.g., *ptc*-GFP, dpERK) in single vs. double mutant backgrounds, "independent" is an overstatement. The term "functionally parallel" would be more appropriate unless cross-regulation is explicitly ruled out.
3. Despite generating a large and high-quality scRNA-seq dataset, the analysis remains descriptive. No trajectory inference or fate-mapping is performed, even though the central claims revolve around hybrid states and differentiation arrest. Methods such as Monocle 3, Palantir, or scVelo could clarify whether mutant cells deviate from canonical lineage trajectories or are blocked at specific transitions.
4. While the co-activation phenotypes resemble dysplasia, the study lacks definitive lineage tracing or clonal analysis to establish tumor origin and progression. The use of broad drivers (e.g., 109-30-Gal4) precludes spatial or temporal precision. Tools such as MARCM, G-TRACE, or temporally restricted Gal80ts could strengthen the argument for cell-autonomous tumor initiation. Furthermore, quantitative assessment of rescue phenotypes (e.g., *zfh1*-RNAi, *pnt*-RNAi) remains incomplete.
5. The conclusion that Hh induces EMT is based solely on transcriptional upregulation of EMT-associated genes. No morphological or functional evidence of EMT, such as polarity loss, cell delamination, or motility, is provided. Immunostaining for polarity markers (e.g., Crb, aPKC), adhesion molecules, and cytoskeletal organization should be used to validate this claim. Without such data, "EMT-like transcriptional state" is the appropriate framing.
6. Mutant cell clusters map poorly to wild-type references, raising concerns about misclassification. Marker misexpression could mimic FSC/pFC expansion. The origin of the tumor-like overgrowth remains unclear. Additional immunostaining, co-expression validation, and lineage tracing are necessary to support the conclusion that tumors arise from pFCs.

7. The authors repeatedly refer to a “hybrid state” to describe cells that co-express markers of both undifferentiated and differentiated follicle cell fates. However, the term “hybrid state” has specific connotations in developmental and cancer biology, typically implying a stable or metastable identity with functional consequences, such as altered plasticity, lineage potential, or behavior. In this study, the designation appears to be based solely on transcriptional co-expression, without supporting evidence for functional hybridity or trajectory disruption. The authors should therefore define more precisely what they mean by “hybrid,” clarify whether this is a transient, transcriptional, or functionally stable state, and consider using a more neutral term (e.g., “mixed transcriptional profile”) unless further dynamic or functional validation is provided.

8. The manuscript surveys several signaling pathways (Wnt, JNK, Hippo), but the main mechanistic insights and tumorigenic phenotypes stem from Hh and EGFR-Ras overactivation. Including additional pathways without comparable depth or functional relevance diffuses the narrative and dilutes the manuscript's impact. The authors should consider narrowing their focus to the Hh and EGFR-Ras axis—which clearly drives the key phenotypes—or explicitly justify the inclusion of other pathways with targeted analyses or integrative comparisons. This would greatly enhance conceptual coherence and mechanistic clarity.

Additional comments:

- The manuscript includes limited pH3 staining to show persistence of mitotic cells beyond stage 6 in EGFR or Ras overactivation conditions, supporting disruption of the mitosis-to-endocycle transition. However, this analysis is qualitative, not systematically quantified across genotypes, and is not extended to the double mutant tumor contexts where cell cycle misregulation is central to the conclusions. Moreover, other key claims, such as accelerated cycling, checkpoint evasion, and altered G1/G2 phase dynamics, are based entirely on transcriptomic inference. These would be significantly strengthened by incorporating functional assays such as EdU/BrdU incorporation for S-phase entry, FACS-based DNA content profiling, and broader use of mitotic and G2/M markers. This would provide crucial validation for the computational predictions and enhance the mechanistic rigor of the study.
- While Gal80ts permits temporal control, the kinetics of tumor initiation and transcriptional changes remain unresolved. A time-course analysis (e.g., staged scRNA-seq or live imaging of pathway reporters) could clarify the sequence and causality of key phenotypic transitions.
- Reference mapping to wild-type atlases is used to support hybrid or misclassified cell identities, but mapping confidence metrics (e.g., Seurat prediction scores or LISI values) are not reported. Including these would improve interpretability and lend quantitative support to claims of lineage disruption.

Minor Points

- Figures are fragmented and key data (e.g., double mutant phenotypes) are scattered; consolidation would aid comprehension.
- Statistical reporting is inconsistent. All n-values and p-values should be included; bar graphs should be replaced with dot/violin plots.
- Methodological transparency should be improved. Full computational parameters and code (e.g., for SCENIC, Milo, batch correction) should be shared via GitHub or Zenodo.
- Terminology such as “tumor” and “independent” should be used with greater precision and caution.
- The discussion of parallels with human cancer should be grounded in conserved pathway logic (e.g., Zfh1/ZEB1, Pnt/ETS) rather than speculation.

Conclusion

This study addresses important questions in epithelial plasticity and oncogenic signaling using a powerful model and modern transcriptomic tools. However, its impact is undermined by conceptual overreach, narrative disorganization, and a lack of mechanistic validation. Major revisions, including trajectory inference, targeted rescue and epistasis experiments, lineage tracing, and clarification of experimental logic, are necessary to realize the full potential of this work. With these improvements, the manuscript would make a valuable contribution to the field.

Reviewer #2

(Remarks to the Author)

The authors use the follicular epithelium of the *Drosophila* ovary to address the contribution of different signaling pathways to cell proliferation and differentiation. They first combine overexpression of the Hh ligand and/or constitutively activated EGFR and Ras1 with single-cell transcriptomics to define the transcriptional profiles of the affected follicular cells. They are able to define several cell clusters based in their transcriptomic profiles, identifying differentially expressed genes downstream of the affected signaling pathways. They also analyze combined over-activation of the same pathways, and identify a highly proliferative phenotype of undifferentiated cells accompanied by novel transcriptomic signatures. Finally, the authors also do a comprehensible phenotypic description of the consequences of their genetic manipulations, as well as epistatic analysis to better define the key components of Hh, EGFR and other signalling pathways involved in the assignation of cell states.

This seems to me as a very significant and timely analysis in which a lot of effort has been put into the comparison between transcriptomic signatures, the action of candidate signaling pathways and the identification of the main components of these pathways through functional genetic analysis. The methodology is aligned with the standards of the field, the description of the results accurate and in my view the results support the many conclusions stated in the manuscript. I am not expert in the

bioinformatic processing of single cell sequencing experiments, and consequently I cannot express a knowledgeable opinion on methods, methodology and data analyses.

In summary, I consider the manuscript a valuable contribution in the fields of cell signaling, cell signaling pathways interactions, regulation of cell differentiation and proliferation and cooperative effects promoting tumoral development that will be of interest to a variety of scientist working in these and related areas, and I am quite happy to recommend its publication. To me the strongest points are: 1) identification of the main components mediating the action of the Hh (zfh1) and EGFR (pointed) signaling pathways, 2) identification of cooperativity in the regulation of gene expression in response to these pathways, 3) critical contribution of the correct timing of EGFR and Hh activation to generate a normal pattern of cellular differentiation, 4) synergistic cellular alterations upon simultaneous activation of the Hh and EGFR pathways. As a mild criticism, I don't really understand the bases of the manuscript title "A single-cell analysis of Hh and EGFR-RAS: Independent signaling pathways prevent tumorigenesis", as it is the fact that signaling pathways during normal development are deployed in an orderly fashion what prevent tumorigenesis. Perhaps the authors would like to give a second thought to make the title more adjusted to the conclusions of the manuscript. Thus, it seems to me that something like "A single-cell analysis of Hh and EGFR-RAS: the timely deployment of these signaling pathways prevent tumorigenesis" could be more adjusted to what they are describing.

Reviewer #3

(Remarks to the Author)

In this ms, Katja Rust and colleagues uses the follicle epithelium of *Drosophila* and combines single cell genomics, bionformatic analysis and functional genetics to identify the transcriptional landscapes driving tumorigenesis. In the first part of the ms, authors identify Zfh1 transcription factor as the key Hh signaling target driving EMT like state. In the second part, they show that Pointed transcription factor mediates the effects of EGFR/Ras on proliferation. In the last part of the ms, authors analyze the effects of co-activation of Hh and EGF/RAs and come to the conclusion that Pointes drives genomic instability. Unfortunately, on one hand, the quality of images to show the effects of all genetic manipulations on cell behavior (proliferation, EMT, etc) is not good. On the other hand, the writing is confusing, hard to follow, and, to a certain extent, devoted to specialists in the field of follicle development rather than to readers more interested in the fundamental consequence of oncogene activation on epithelial tissues. The main message of the paper is indeed unclear. Most experiments are based on overexpression experiments and many claims (eg. Pointed mediates chromosomal instability downstream of EGF-Ras) are not substantiated by bona fide epistatic analysis. Many other conclusions are just based on correlations. Overall, the ms is in a highly preliminary state.

Reviewer #4

(Remarks to the Author)

Rust et al. investigate the consequences of pro-proliferative pathway overexpression on tissue homeostasis using the *Drosophila* follicular epithelium as model system. This epithelial tissue contains stem cells and is continuously developing throughout female adult life to produce egg chambers. Proliferation and differentiation of follicle cells are tightly regulated by multiple signaling pathways, including EGFR-Ras and Hedgehog (Hh). This work brings novelty in using single cell RNA-sequencing approaches to compare the transcriptional landscapes resulting from the targeting of several pro-proliferative pathways in the follicle cell lineage. The authors propose that either Hh or EGFR-Ras overactivation leads to a precursor stage of tumor development while the co-activation of those two pathways creates a synergistic effect and favors the progression towards an advanced cancer stage. The study is strengthened by convincing *in vivo* assays to assess the morphological and physiological consequences of such perturbations. This solid work brings new insights into signaling pathway interactions within a specific tissue and should have a strong impact in the field of tissue homeostasis and cancer development. Nevertheless, the following points should be addressed by the authors to improve clarity of the manuscript and solidity of the results:

Major points

- 1) The result section of the manuscript is quite complex and difficult to read. As it stands, it is restricted to single cell transcriptomic algorithm users. To be addressed to a more general audience and gain in attractivity, the author should provide a more didactic version of the manuscript. Some aspects of the methodology and data representation are not explained at all, nor in the main text nor in the legends. This is the case in particular for the use of milo analysis to provide differential abundance plots and neighborhood graphs.
- 2) While the data seems solid and the analysis well done, the authors have not made the code available. This is an integral part of the paper and access to the code should be provided.
- 3) The role of Rasv12 and top4.4 in polar cell differentiation is not clear. The authors show that polar cell fate is affected upon Rasv12 expression in polar cells, as seen by the downregulation of *dpn*, *upd1* and *upd3* expression (Supplementary Figure 4L). This is also supported by the decrease of Dpn detection in Rasv12 follicles (Supplementary 5B). These results suggest that Ras signaling inhibits polar cell differentiation rather than promoting it as the authors conclude. The role of top4.4 in polar cell differentiation is not very clear since *dpn* expression seems lower than in control but higher than upon Rasv12 expression. Also, are Dpn protein levels higher in top4.4 follicles than in controls (there is no control in Supplementary Figure 5A,B). Finally, there are more polar cells per cluster upon Rasv12 expression but there is no biological conclusion drawn from this result. Since polar cells are found in excess in early follicle stages and are eliminated by apoptosis before stage 5, the extra polar cells per cluster phenotype could result from an immature differentiation state or from an apoptosis downregulation. It would be interesting to look at apoptotic markers in this condition and more generally to document the status of apoptosis in Rasv12 follicles?

4) The authors show that sustained EGFR-Ras overactivation blocks the entry of MB follicle cells to endocycles and they suggest that this effect is mediated by pnt. However, the demonstration is rather correlative with a phenocopy of pnt and Rasv12 overexpression phenotypes. A decrease in pnt expression in Rasv12 follicles should be made to reach that conclusion. In addition, it is already known that pnt is a target of EGFR signaling, hence the sentence "(...) ovarioles with pnt overexpression contained mitotic cells across all stages, suggesting that pnt is a key target of the EGFR-Ras pathway (Fig. 4J-K)" cannot be written.

Minor points

- 1) Regarding the fluidity of the manuscript, it is difficult to follow since the authors examine many different aspects that are not always linked and some conclusions do not fit easily with the general message. For instance, the second paragraph of the first part of the result section is fairly substantial in focusing on new insights provided by the single cell RNA-seq performed in the wild type condition, which does not fit with the title of this section. While the content of this paragraph is interesting, it is not connected to the rest of the section and distracts the reader from the main focus.
- 2) The authors examine the mitosis to endocycle transition, which occurs at stage 6, in different genetic contexts. They observe PH3 staining in the latest follicles upon Rasv12 expression, as opposed to controls, but it could be because follicles are maintained in an early stage and thus do not reach the switch to endocycling. How were follicle stages assessed in Rasv12 follicles, given that regular criteria for staging, such as follicle size and shape are lost? Along this line, FC 6+ cells are largely under-represented in the Rasv12 single cell transcriptome.
- 3) The authors show that pnt overexpression in FCs blocks the switch to endocycles. However, posterior follicle cells are known to express pnt following EGFR activation beyond stage 6, but they nevertheless switch to endocycling. It would be interesting to comment on that difference.
- 4) The authors mention in the text some Zfh1+ and Eya+ cells in reference to Figure 5D-E, but Eya staining is not shown in the figure.
- 5) There is a reference to Figure S6K in the text but Figure S6 stops with the H panel. In addition, the figure legend goes up to panel L.

Reviewer #5

(Remarks to the Author)

Version 1:

Reviewer comments:

Reviewer #1

(Remarks to the Author)

This revised manuscript represents a substantial and thoughtful improvement over the previous version. The authors have clearly invested significant effort in addressing both conceptual and technical concerns, particularly by restructuring the Results, narrowing the focus to Hh and EGFR-Ras signaling, and adding extensive new experimental data.

The additional epistasis experiments and quantitative pathway reporter analyses considerably strengthen the claim that Hh and EGFR-Ras do not cross-regulate each other at the level of pathway activation. Similarly, the inclusion of pseudotime analyses, mapping confidence scores, staged phenotypic analyses, and quantitative rescue experiments markedly improves the rigor of the single-cell and genetic conclusions. The new data supporting an EMT-like phenotype downstream of Hh/Zfh1—combining transcriptional signatures with polarity defects and live imaging—are particularly compelling and elevate the mechanistic depth of the study.

That said, a few conceptual vulnerabilities remain. Most notably, the manuscript continues to use the term "independent" in a way that may be interpreted more broadly than warranted by the data. While pathway activation appears independent, the strong transcriptional convergence and cooperative effects on proliferation and differentiation suggest functional convergence at downstream regulatory nodes. A slightly more cautious framing—especially in the Abstract and Discussion—would reduce the risk of overstatement.

Likewise, although the evidence for tumor-like growth upon co-activation of Hh and EGFR-Ras is strong within the *Drosophila* ovary context, the use of the term "tumor" may still raise concerns for some readers. An explicit definition of what constitutes a tumor in this system, and a brief acknowledgment of which classical cancer criteria are and are not met, would improve clarity and pre-empt criticism.

The identification of MB cells as the likely cell of origin for the overgrowth phenotype is a clear advance, enabled by the use of additional cell-type-specific Gal4 drivers. However, because this conclusion relies on driver specificity rather than lineage tracing, the inherent limitations of this approach should be stated explicitly in the manuscript itself.

Finally, while the single-cell analyses are comprehensive and technically sound, the Results section remains analytically dense. Select consolidation of scRNA-seq figures—without reducing data availability—could further sharpen the central biological narrative.

In summary, this is now a strong and carefully executed study that makes a meaningful contribution to our understanding of how parallel signaling pathways jointly constrain epithelial homeostasis. With modest adjustments to terminology, framing, and emphasis, the manuscript should be well positioned for acceptance.

Reviewer #3

(Remarks to the Author)

Reviewer #4

(Remarks to the Author)

The authors have drastically restructured the manuscript and restricted the scope of the study, which now focuses on Hh and EGFR signaling pathways, leading to an improved revised version that gained in clarity and interest. The authors have also made substantial efforts to add convincing new data and provide more detailed explanations of experimental procedures, therefore addressing all our concerns.

Nevertheless, a couple of minor points should be addressed:

- We agree with Reviewer 2 that the title of the manuscript does not reflect correctly the results of the study. That Hh and EGFR pathways induce tumorigenesis when overactivated does not mean that their physiological role is to prevent tumorigenesis. It is quite the opposite that would be required to be able to draw that conclusion, i.e if a loss of pathway activity would lead to tumorigenesis. As it stands the title (but also some statements in the abstract and other parts of the manuscript) is misleading. The title is only true in the context of overactivation. In addition, it suggests that there is a mechanism in place to limit pathway activity, hence preventing tumorigenesis, but this is not investigated in this work. Same for the following section title: "Independent regulatory modes of proliferation prevents expansion of undifferentiated cells", this is true upon pathway overactivation. Along the same line, why the authors undertook a gain-of-function approach in the first place is not clear.

- Also in the section "Independent regulatory modes of proliferation prevents expansion of undifferentiated cells", the conclusion « co-overactivation of independent signaling pathways protects the tissue from aberrant expansion of undifferentiated cells and therefore represents a safety mechanism for epithelial homeostasis » is wrong.

- What are stalk cells in EGFRCA and RasV12 since follicles are fused?

- Panels C and Y from Figure 5 are very difficult to read with the color code used. Comparison of average expression in the different conditions with wild-type is not possible.

Reviewer #5

(Remarks to the Author)

RESPONSE TO REVIEWER COMMENTS

Reviewer #1 (Remarks to the Author):

Reviewer Comments on “A single-cell analysis of Hh and EGFR-Ras: Independent signaling pathways prevent tumorigenesis”

Rust et al. present a technically sophisticated study combining genetic perturbation and single-cell transcriptomics to explore how Hh and EGFR-Ras signaling affect cell fate and proliferation in the *Drosophila* follicle epithelium. The authors identify transcriptional hybrid states, altered cell cycle control, and overgrowth phenotypes under pathway co-activation, which they interpret as hallmarks of tumorigenesis. While the dataset is rich and the experimental system well-chosen, the study’s interpretive clarity and mechanistic depth are hampered by key conceptual, analytical, and structural weaknesses.

Major Points

1. The manuscript suffers from a lack of a clear conceptual thread. The experimental flow often feels observational rather than hypothesis-driven, with new datasets introduced without clear rationale. This impairs readability and interpretability, even for domain experts. To improve clarity and engagement, the authors should reorganize the results around discrete, testable questions and explicitly state the purpose of each experimental step.

We thank the reviewer for the suggestion. We have restructured the results section to create a more streamlined manuscript and focused on our key data. Specifically, we now concentrated on the Hh, EGFR, Ras and co-overactivation datasets and reorganized the results, starting each section with a clear question or aim.

2. The assertion that Hh and EGFR-Ras function independently is not supported by mechanistic data. The pathways converge on shared transcriptional targets and exhibit synergistic phenotypes. Without epistasis experiments or systematic pathway activity measurements (e.g., ptc-GFP, dpERK) in single vs. double mutant backgrounds, “independent” is an overstatement. The term “functionally parallel” would be more appropriate unless cross-regulation is explicitly ruled out.

We agree with the reviewer’s comment and therefore addressed pathway independence experimentally (Fig. 1 and S1). Our quantifications of pathway reporter activity and epistasis experiments suggest that the pathways are indeed independent. We added the following section in the results: “EGFR-Ras signaling and the Hh pathway act independently in the follicle epithelium”.

We also investigated whether reduction of either pathway in the double overactivation background restores the respective phenotypes, although we did not include this data in the manuscript. This experiment, too, suggests that both pathways are independent (see attached figure, $n = 82$ (Raf-RNAi), 70 (smo-RNAi)).

We further measured pathway activity in the co-overactivation background and found similar levels (6 days post temperature shift: dpERK: EGFR^{CA} vs Hh^{OE},EGFR^{CA}: $p = 0.1441$ (ns), Ras^{V12} vs Hh^{OE}, Ras^{V12}: $p = 0.4652$ (ns), ptc-cherry: Hh^{OE} vs Hh^{OE},EGFR^{CA}: $p = 0.1985$ (ns), vs Hh^{OE},Ras^{V12}: $p = 0.03212$ (*), p -values from Kruskal-Wallis tests). Together, these results support that no direct cross-regulation between the Hh and EGFR-Ras pathways takes place.

Ovarioles expressing Hh^{OE}, Ras^{V12} under the *109-30* promoter combined with *Raf*-RNAi (left) or *smo*-RNAi (right) and stained for DAPI (blue), *Eya* (green) and *ECad* (magenta). Depletion of *Raf* restores a Hh^{OE} -like phenotype with large stalk-like areas and lacking MB cells from mature cysts. Depletion of *smo* restores a Ras^{V12} -like phenotype with cyst fusions. Note that we temperature-shifted for 6 days.

3. Despite generating a large and high-quality scRNA-seq dataset, the analysis remains descriptive. No trajectory inference or fate-mapping is performed, even though the central claims revolve around hybrid states and differentiation arrest. Methods such as Monocle 3, Palantir, or scVelo could clarify whether mutant cells deviate from canonical lineage trajectories or are blocked at specific transitions.

We performed Monocle3 analysis and found that the results indeed support our claims (see Fig. 2,6, S4 and S0). We thank the reviewer for this fruitful suggestion.

4. While the co-activation phenotypes resemble dysplasia, the study lacks definitive lineage tracing or clonal analysis to establish tumor origin and progression. The use of broad drivers (e.g., 109-30-Gal4) precludes spatial or temporal precision. Tools such as MARCM, G-TRACE, or temporally restricted Gal80ts could strengthen the argument for cell-autonomous tumor initiation.

We performed two sets of experiments to address these concerns:

- We performed staged analysis of tumor growth (Fig. S12)
- We applied several cell-type specific Gal4 drivers, including an additional driver, *zfh1*-Gal4, which allowed us to conclude that the tumor origin is the MB follicle cell population (Fig. S11)

From our results we conclude that the tumor-initiation depends on Gal4 activity in MB cells: “To test whether pFCs indeed give rise to this expansion of cells, we made use of Gal4 driver lines^{12,60} targeting distinct follicle cell populations to express Hh^{OE} with either $EGFR^{CA}$ or Ras^{V12} (Fig. S11A-F). We found that only Gal4 drivers with expression in MB cells caused tumor-like growth (Fig. S11G-N). This demonstrates that the aberrant expansion of Hh - $EGFR/Ras$ cells is likely caused by MB follicle cells retaining pFC character.”

Furthermore, quantitative assessment of rescue phenotypes (e.g., *zfh1*-RNAi, *pnt*-RNAi) remains incomplete.

We added quantifications for all phenotypes described in the paper. The quantifications of *zfh1*- and *pnt*-RNAis, specifically, can be found in Fig. 5Y and S6U.

5. The conclusion that Hh induces EMT is based solely on transcriptional upregulation of EMT-associated genes. No morphological or functional evidence of EMT, such as polarity loss, cell delamination, or motility, is provided. Immunostaining for polarity markers (e.g., *Crb*, *aPKC*), adhesion molecules, and cytoskeletal organization should be used to validate this claim. Without such data, “EMT-like transcriptional state” is the appropriate framing.

We investigated the EMT phenotype in more depth, as suggested and summarized our results in the paragraph “Ectopic Hh signaling induces EMT” in the Results section and in Figure 4. We now show that Hh overactivation induces Pbl

and Htl using reporter lines. We also investigated cell polarity and confirmed that Hh overactivation induces polarity defects. Further, we applied live imaging analysis and found migration effects and evidence that Hh overactivated cells display invasive migration. Please refer to Fig. 4 and Movies S1-3.

6. Mutant cell clusters map poorly to wild-type references, raising concerns about misclassification. Marker misexpression could mimic FSC/pFC expansion. The origin of the tumor-like overgrowth remains unclear. Additional immunostaining, co-expression validation, and lineage tracing are necessary to support the conclusion that tumors arise from pFCs.

We added an additional cell-type specific Gal4 driver line, *zfh1-Gal4*, which allows us to discriminate between the pFCs and MB cells as tumor origin. Indeed, we find that MB cells and not pFCs are the cell type that induces tumorigenic growth (Fig. S11).

Please note that we considered performing lineage tracing analysis, but the only cell type that can be reliably traced in the ovary is the FSC, as all other cells are continuously replaced. As our data show that Gal4 activity in FSCs does not induce tumor growth, we therefore refrained from performing lineage tracing.

In addition, we performed pseudotime analysis, which supports our initial claim that follicle cells with Hh-EGFR/Ras co-overactivation do not differentiate beyond the initial stages of follicle cell development (Fig. 6O and S4D-G). We also performed an additional immunostaining analysis of BrC, which is expressed from Stage 6 of WT MB differentiation (Fig. S8F). We find that follicle cells with Hh-EGFR/Ras co-overactivation do not induce the expression of BrC, except in very few cases in which BrC positive cells were associated with germline cysts with WT-like morphology. We added this information in the manuscript: “These cells did not induce the expression of BrC, a marker of Stage 6 MB cells, except in very rare cases (data not shown).” Please refer to the figure below.

Ovarioles expressing Hh^{OE} and either EGFR^{CA} (upper panel) or Ras^{V12} (lower panel) stained for DAPI (blue), BrC (green) and pH3 (magenta). We occasionally detected BrC positive cells (green line), which were associated with germline cysts.

7. The authors repeatedly refer to a “hybrid state” to describe cells that co-express markers of both undifferentiated and differentiated follicle cell fates. However, the term “hybrid state” has specific connotations in developmental and cancer biology, typically implying a stable or metastable identity with functional consequences, such as altered plasticity, lineage potential, or behavior. In this study, the designation appears to be based solely on transcriptional co-expression, without supporting evidence for

functional hybridity or trajectory disruption. The authors should therefore define more precisely what they mean by “hybrid,” clarify whether this is a transient, transcriptional, or functionally stable state, and consider using a more neutral term (e.g., “mixed transcriptional profile”) unless further dynamic or functional validation is provided.

As suggested, we now refer to this state as a “mixed transcriptional state”. We do raise the reader’s attention to a similar “hybrid state” in cancer metastasis in the discussion.

8. The manuscript surveys several signaling pathways (Wnt, JNK, Hippo), but the main mechanistic insights and tumorigenic phenotypes stem from Hh and EGFR-Ras overactivation. Including additional pathways without comparable depth or functional relevance diffuses the narrative and dilutes the manuscript's impact. The authors should consider narrowing their focus to the Hh and EGFR-Ras axis—which clearly drives the key phenotypes—or explicitly justify the inclusion of other pathways with targeted analyses or integrative comparisons. This would greatly enhance conceptual coherence and mechanistic clarity.

As suggested, we removed the Wnt, yki and Hep datasets and now focus on the Hh, EGFR, Ras and co-overactivation datasets.

Additional comments:

The manuscript includes limited pH3 staining to show persistence of mitotic cells beyond stage 6 in EGFR or Ras overactivation conditions, supporting disruption of the mitosis-to-endocycle transition. However, this analysis is qualitative, not systematically quantified across genotypes, and is not extended to the double mutant tumor contexts where cell cycle misregulation is central to the conclusions.

We quantified phospho-Histone H3 staining across genotypes (Fig. 5O) and investigated phospho-Histone H3 in the co-overactivation background (Fig. 7L-M). Each ovariole in the co-overactivation background we examined showed the phenotype shown in Fig. 7L-M (n = 34 ovarioles for each condition).

Moreover, other key claims, such as accelerated cycling, checkpoint evasion, and altered G1/G2 phase dynamics, are based entirely on transcriptomic inference. These would be significantly strengthened by incorporating functional assays such as EdU/BrdU incorporation for S-phase entry, FACS-based DNA content profiling, and broader use of mitotic and G2/M markers. This would provide crucial validation for the computational predictions and enhance the mechanistic rigor of the study.

We concentrated on a more detailed analysis of the cell cycle. For this, we used phospho-Histone H3 staining and quantification and utilized the CycB-nls-RFP construct of the Fly-FUCCI system, which allowed us to identify cells in M and G1, respectively (Fig. 5I-O, 7L-S, S7B-G). We were able to confirm our cell cycle scoring results in the wildtype background (Fig. SB). Further, we show and quantify cell cycle defects in single and double overactivation backgrounds (Fig. 5I-O, 7L-S & S7C-G).

- While Gal80ts permits temporal control, the kinetics of tumor initiation and transcriptional changes remain unresolved. A time-course analysis (e.g., staged scRNA-seq or live imaging of pathway reporters) could clarify the sequence and causality of key phenotypic transitions.

We performed staged analysis of double overactivation phenotypes and combined them with pathway reporter analysis (Fig. S12). Our results show that pathway activity is induced shortly after temperature shift, in agreement with the timing of Gal4 system induction and then stays consistently active while the phenotypic alteration continuously increases with time.

- Reference mapping to wild-type atlases is used to support hybrid or misclassified cell identities, but mapping confidence metrics (e.g., Seurat prediction scores or LISI values) are not reported. Including these would improve interpretability and lend quantitative support to claims of lineage disruption.

We included mapping scores for each dataset in Fig. 2G (as a Dotplot representation), S4H-K (on UMAP plots) and S10F-G (Dotplot and UMAP plots).

Minor Points

- **Figures are fragmented and key data (e.g., double mutant phenotypes) are scattered; consolidation would aid comprehension.**

We reorganized the results section and organized data in a more streamlined manner.

- **Statistical reporting is inconsistent. All n-values and p-values should be included; bar graphs should be replaced with dot/violin plots.**

All barplots have been replaced by violin plots. We also report all n and p-values as well as the statistical test applied for each calculation in the figure legends.

- **Methodological transparency should be improved. Full computational parameters and code (e.g., for SCENIC, Milo, batch correction) should be shared via GitHub or Zenodo.**

We now provide all scripts under the following Github page:

<https://github.com/KatjaRM/Anschuetz-SCS-Hh-and-EGFR-Ras>

- **Terminology such as “tumor and “independent” should be used with greater precision and caution.**

We agree with the reviewer. We now provide extensive experimental evidence that both pathways act independently (Fig. 1, S1D-M) and are therefore confident in the use of this term. We further assess several cancer hallmarks in the double overactivation backgrounds that support our conclusion that this co-overactivation results in tumors (Fig. 7K, S12, S13). In addition, we revised the text and use both terms only when mandated.

- **The discussion of parallels with human cancer should be grounded in conserved pathway logic (e.g., Zfh1/ZEB1, Pnt/ETS) rather than speculation.**

We revised the discussion accordingly and added the following paragraph in the discussion: “In human cancer, the Zfh1 homologs ZEB1/2 are relevant target genes downstream of Hh signaling, where they promote EMT and therefore metastasis ⁷⁷. Similarly, the Pnt homologs ETS1/2 are established targets of the MAPK signaling pathway in cancer and promote tumor proliferation and invasiveness ⁷⁸. This highlights the high conservation of the Hh and EGFR-Ras signaling pathways.”

Conclusion

This study addresses important questions in epithelial plasticity and oncogenic signaling using a powerful model and modern transcriptomic tools. However, its impact is undermined by conceptual overreach, narrative disorganization, and a lack of mechanistic validation. Major revisions, including trajectory inference, targeted rescue and epistasis experiments, lineage tracing, and clarification of experimental logic, are necessary to realize the full potential of this work. With these improvements, the manuscript would make a valuable contribution to the field.

We thank the reviewer for their valuable comments. We believe that addressing each concern has strongly improved the manuscript and are grateful for the time and effort the reviewer spent on revising our manuscript. We are confident that we have addressed all comments comprehensively and hope that this erases the concerns.

Reviewer #2 (Remarks to the Author):

The authors use the follicular epithelium of the Drosophila ovary to address the contribution of different signaling pathways to cell proliferation and differentiation. They first combine overexpression of the Hh ligand

and/or constitutively activated EGFR and Ras1 with single-cell transcriptomics to define the transcriptional profiles of the affected follicular cells. They are able to define several cell clusters based in their transcriptomic profiles, identifying differentially expressed genes downstream of the affected signaling pathways. They also analyze combined over-activation of the same pathways, and identify a highly proliferative phenotype of undifferentiated cells accompanied by novel transcriptomic signatures. Finally, the authors also do a comprehensible phenotypic description of the consequences of their genetic manipulations, as well as epistatic analysis to better define the key components of Hh, EGFR and other signalling pathways involved in the assignation of cell states.

This seems to me as a very significant and timely analysis in which a lot of effort has been put into the comparison between transcriptomic signatures, the action of candidate signaling pathways and the identification of the main components of these pathways through functional genetic analysis. The methodology is aligned with the standards of the field, the description of the results accurate and in my view the results support the many conclusions stated in the manuscript. I am not expert in the bioinformatic processing of single cell sequencing experiments, and consequently I cannot express a knowledgeable opinion on methods, methodology and data analyses.

In summary, I consider the manuscript a valuable contribution in the fields of cell signaling, cell signaling pathways interactions, regulation of cell differentiation and proliferation and cooperative effects promoting tumoral development that will be of interest to a variety of scientist working in these and related areas, and I am quite happy to recommend its publication. To me the strongest points are: 1) identification of the main components mediating the action of the Hh (zfh1) and EGFR (pointed) signaling pathways, 2) identification of cooperativity in the regulation of gene expression in response to these pathways, 3) critical contribution of the correct timing of EGFR and Hh activation to generate a normal pattern of cellular differentiation, 4) synergistic cellular alterations upon simultaneous activation of the Hh and EGFR pathways.

We thank the reviewer for their time and effort and their input.

As a mild criticism, I don't really understand the bases of the manuscript title "A single-cell analysis of Hh and EGFR-RAS: Independent signaling pathways prevent tumorigenesis", as it is the fact that signaling pathways during normal development are deployed in an orderly fashion what prevent tumorigenesis. Perhaps the authors would like to give a second thought to make the title more adjusted to the conclusions of the manuscript. Thus, it seems to me that something like "A single-cell analysis of Hh and EGFR-RAS: the timely deployment of these signaling pathways prevent tumorigenesis" could be more adjusted to what they are describing.

We agree that the previous version of the manuscript did not address the independence of the signaling pathways and therefore addressed this question experimentally. We analyzed pathway activity reporters and performed epistasis experiments which suggest that, indeed, the Hh and EGFR-Ras pathway act independent in follicle cells (Fig. 1, S1D-M). We therefore chose the following, revised title for our manuscript: "Independent Hh and EGFR-Ras pathway activity sustains epithelial homeostasis and prevents tumorigenesis"

Reviewer #3 (Remarks to the Author):

In this ms, Katja Rust and colleagues uses the follicle epithelium of *Drosophila* and combines single cell genomics, bionformatic analysis and functional genetics to identify the transcriptional landscapes driving tumorigenesis. In the first part of the ms, authors identify Zfh1 transcription factor as the key Hh signaling target driving EMT like state. In the second part, they show that Pointed transcription factor mediates the effects of EGFR/Ras on proliferation. In the last part of the ms, authors analyze the effects of co-activation of Hh and EGF/RAs and come to the conclusion that Pointes drives genomic instability. Unfortunately, on one hand, the quality of images to show the effects of all genetic manipulations on cell behavior (proliferation, EMT, etc) is not good. On the other hand, the writing is confusing, hard to follow, and, to a certain extent, devoted to specialists in the field of follicle development rather than to readers more interested in the

fundamental consequence of oncogene activation on epithelial tissues. The main message of the paper is indeed unclear. Most experiments are based on overexpression experiments and many claims (eg. Pointed mediates chromosomal instability downstream of EGF-Ras) are not substantiated by bona fide epistatic analysis. Many other conclusions are just based on correlations. Overall, the ms is in a highly preliminary state.

We thank the reviewer for their honest analysis and agree that the structure of the results section of our manuscript required fundamental reorganization. We have therefore:

- Concentrated on the key findings in the new manuscript and removed datasets that we did not analyse in depth
- Optimized writing in order to make the manuscript accessible to a broad audience
- Performed diverse experiments that confirm that the Hh and EGFR-Ras pathways act independently in early follicle epithelial homeostasis (Fig. 1, S1D-M)
- Added in-depth analysis of EMT induction (Fig. 4 and Movies S1-3)
- Performed extensive analysis of cell cycle defects (Fig. 5G-O, 7L-S, S7B-G, S13)
- Updated images of phenotypes and added extensive quantifications

Reviewer #4 (Remarks to the Author):

Rust et al. investigate the consequences of pro-proliferative pathway overexpression on tissue homeostasis using the *Drosophila* follicular epithelium as model system. This epithelial tissue contains stem cells and is continuously developing throughout female adult life to produce egg chambers. Proliferation and differentiation of follicle cells are tightly regulated by multiple signaling pathways, including EGFR-Ras and Hedgehog (Hh). This work brings novelty in using single cell RNA-sequencing approaches to compare the transcriptional landscapes resulting from the targeting of several pro-proliferative pathways in the follicle cell lineage. The authors propose that either Hh or EGFR-Ras overactivation leads to a precursor stage of tumor development while the co-activation of those two pathways creates a synergistic effect and favors the progression towards an advanced cancer stage. The study is strengthened by convincing *in vivo* assays to assess the morphological and physiological consequences of such perturbations. This solid work brings new insights into signaling pathway interactions within a specific tissue and should have a strong impact in the field of tissue homeostasis and cancer development.

We thank the reviewer for their analysis, effort and time.

Nevertheless, the following points should be addressed by the authors to improve clarity of the manuscript and solidity of the results:

Major points

1) The result section of the manuscript is quite complex and difficult to read. As it stands, it is restricted to single cell transcriptomic algorithm users. To be addressed to a more general audience and gain in attractivity, the author should provide a more didactic version of the manuscript. Some aspects of the methodology and data representation are not explained at all, nor in the main text nor in the legends. This is the case in particular for the use of milo analysis to provide differential abundance plots and neighborhood graphs.

We revised the results section and reorganized it entirely. We now concentrate on our key results and present all data in a more accessible manner. We also provide a more detailed explanation of the milo approach: "To assess the effects that either signaling pathway had on each cell type, we performed differential abundance testing, which tests whether specific cell types or states are over- or underrepresented between conditions ³⁷." Further, we do not discuss Neighborhood graphs in depth in the revised manuscript, although we still provide this data in the supplementary information (Fig. S4A-C, S10I).

2) While the data seems solid and the analysis well done, the authors have not made the code available. This is an integral part of the paper and access to the code should be provided.

All scripts are now publicly accessible under <https://github.com/KatjaRM/Anschuetz-SCS-Hh-and-EGFR-Ras>.

3) The role of Rasv12 and top4.4 in polar cell differentiation is not clear. The authors show that polar cell fate is affected upon Rasv12 expression in polar cells, as seen by the downregulation of *dpn*, *upd1* and *upd3* expression (Supplementary Figure 4L). This is also supported by the decrease of Dpn detection in Rasv12 follicles (Supplementary 5B). These results suggest that Ras signaling inhibits polar cell differentiation rather than promoting it as the authors conclude. The role of top4.4 in polar cell differentiation is not very clear since *dpn* expression seems lower than in control but higher than upon Rasv12 expression. Also, are Dpn protein levels higher in top4.4 follicles than in controls (there is no control in Supplementary Figure 5A,B). Finally, there are more polar cells per cluster upon Rasv12 expression but there is no biological conclusion drawn from this result. Since polar cells are found in excess in early follicle stages and are eliminated by apoptosis before stage 5, the extra polar cells per cluster phenotype could result from an immature differentiation state or from an apoptosis downregulation. It would be interesting to look at apoptotic markers in this condition and more generally to document the status of apoptosis in Rasv12 follicles?

In response to this comment, we analyzed the polar cell specific effects of Ras^{V12} further and found that, in contrast to Ltop^{4.4}, Ras^{V12} induces ectopic mitosis in polar cells (Fig. S5I-M). We also depleted MAPK activity in polar cells using two transgenic lines (ERK^{DN} and *Raf*-RNAi), which did not affect polar cell numbers (Fig. S5H). Additionally, our analysis of pathway activity indicates that Ras^{V12} has a substantially higher effect on MAPK activity. Together, we conclude that Ras^{V12} induces high levels of MAPK activity which results in ectopic mitosis in polar cells, causing the increased polar cell number: "To test whether Ras^{V12} indeed exerts effects on polar cells distinct from those induced by EGFR^{CA} we examined polar cells further. Polar cell specific induction of EGFR^{CA} using *upd*-Gal4 did not result in any morphological defects when compared to a wildtype control (Fig. S5E-F). In contrast, polar cell specific Ras^{V12} expression significantly increased the number of polar cells per cluster and caused a double row stalk phenotype (Fig. S5G-H). Polar cell specific expression of ERK^{DN} or *Raf*-RNAi did not alter polar cell numbers, suggesting that the MAPK-pathway is not required to be active in polar cells for their differentiation (Fig. S5H). However, we noted that Ras^{V12} polar cells were occasionally mitotic (as indicated by staining for the mitotic marker, phospho-Histone H3), while we did not observe mitotic control and EGFR^{CA} polar cells, in agreement with previous reports ⁴¹ (Fig. S5I-M). As induction of proliferation is a well-described effect of the EGFR-Ras pathway and in agreement with our analysis on the pathway reporter dpERK, we conclude that the Ras^{V12} transgene induces MAPK signaling more effectively than EGFR^{CA}, resulting in ectopic mitotic divisions of polar cells."

Our results are in line with an interpretation where Ras^{V12} is able to induce higher levels of MAPK activity than EGFR^{CA}, which causes an aberrant expansion of the polar cell pool. Prolonged mitosis blocks differentiation in many developmental contexts. Therefore, it is tempting to speculate that ectopic mitosis in early stages of polar cell-fated pFCs induced by Ras^{V12} may block polar cell differentiation. We attempted to stain for Dcp1 in ovarioles with expression of Ras^{V12} specifically in polar cells with *upd*-Gal4 but very few flies of the appropriate genotype reach adulthood, so we could not obtain a sufficient sample size for a conclusive analysis. We instead decided to use the limited number of flies we obtained to complete the high priority goal of quantifying the mitosis defects. As the effect of Ras^{V12} signaling on polar cell differentiation is not the main focus of this manuscript, a comprehensive investigation of this question is beyond the scope of this work and would be more appropriately addressed in a follow-up study..

4) The authors show that sustained EGFR-Ras overactivation blocks the entry of MB follicle cells to endocycles and they suggest that this effect is mediated by *pnt*. However, the demonstration is rather correlative with a phenocopy of *pnt* and Rasv12 overexpression phenotypes. A decrease in *pnt* expression in Rasv12 follicles should be made to reach that conclusion. In addition, it is already known that *pnt* is a target of EGFR signaling, hence the sentence "(...) ovarioles with *pnt* overexpression contained mitotic cells across all stages, suggesting that *pnt* is a key target of the EGFR-Ras pathway (Fig. 4J-K)" cannot be written.

We agree with the reviewer and investigated mitosis in the background of EGFR and Ras overactivation combined with *pnt*-RNAi. Indeed, this did not fully rescue ectopic mitosis (Fig. S7H-J). We therefore now conclude that *pnt* is not the only crucial target gene of EGFR-Ras signaling: "Hence, *pnt* is one, but likely not the only, important target of EGFR-Ras signaling in this process."

We reference publications in the manuscript identifying Pnt as a Ras target gene and have removed the previous sentence. Eg. "E2f1 is a well-known cell cycle regulator required for G1 to S transition and Pnt induces cell cycle related genes during neural development and is induced by Ras signaling⁴⁶⁻⁴⁸."

Minor points

1) Regarding the fluidity of the manuscript, it is difficult to follow since the authors examine many different aspects that are not always linked and some conclusions do not fit easily with the general message. For instance, the second paragraph of the first part of the result section is fairly substantial in focusing on new insights provided by the single cell RNA-seq performed in the wild type condition, which does not fit with the title of this section. While the content of this paragraph is interesting, it is not connected to the rest of the section and distracts the reader from the main focus.

We reorganized the results section, which is now more streamlined.

2) The authors examine the mitosis to endocycle transition, which occurs at stage 6, in different genetic contexts. They observe PH3 staining in the latest follicles upon Rasv12 expression, as opposed to controls, but it could be because follicles are maintained in an early stage and thus do not reach the switch to endocycling. How were follicle stages assessed in Rasv12 follicles, given that regular criteria for staging, such as follicle size and shape are lost? Along this line, FC 6+ cells are largely under-represented in the Rasv12 single cell transcriptome.

Correct, late stage follicle cells are underrepresented in the Ras^{V12} as well as EGFR^{CA} datasets (Fig. 2B-C). In Ras^{V12} ovarioles this is also reflected in the phenotype. We note in the manuscript: "In ovarioles with overexpression of Ras^{V12}, the normal structure of the ovariole was completely disrupted and follicles rarely formed at all (Fig. 1F)."

We attempted to use differentiation markers (including BrC and *eya*) to stage follicle cells but found that these markers cannot provide sufficient resolution to time postmitotic follicle cells in the wildtype context (see Fig. S8A and F). Therefore, we instead staged follicles based on DAPI morphology of nurse cells as previously described (Jia et al., 2016 DOI: 10.1038/srep18850) and focused our analysis on MB cells associated with these cysts.

To address whether MB follicle cells associated with late stage germline cysts differentiate, we analyzed *eya* and BrC expression. In WT ovarioles, Stages shortly after mitosis downregulate *Eya* and BrC is expressed shortly before MB cells exit mitosis (Fig. S8). We found that Hh overactivation expands *eya* expression and identified ectopic Hh^{OE} mitotic cells, which are *Eya* positive (Fig. S8). At the same time, BrC expression is only induced in a few Hh^{OE} MB cells, suggesting that ectopic mitosis induced by Hh^{OE} is a consequence of delayed differentiation. In contrast, the expression status of *eya* and BrC in EGFR-Ras overactivated MB cells displaying ectopic mitosis suggests that these cells are competent to reach a differentiation status equivalent to endocycling MB cells (Fig. S8).

3) The authors show that pnt overexpression in FCs blocks the switch to endocycles. However, posterior follicle cells are known to express pnt following EGFR activation beyond stage 6, but they nevertheless switch to endocycling. It would be interesting to comment on that difference.

We thank the reviewer for raising this important point and now address it in the discussion: "Notably, Pnt is known to be induced by EGFR signaling in posterior follicle cells past Stage 6, where it promotes the expression of midline to mediate the posterior follicle cell fate and hence actively promotes differentiation of MB cells that have already entered the endocycle⁷⁸. This indicates that, while EGFR-Ras signaling must be inactive for cells to enter endocycling, activation of EGFR-Ras signaling postmitotic MB is insufficient to induce reentry into mitosis. Future studies will have to address whether this depends on the exact timing or magnitude of EGFR-Ras activity and which target genes besides pnt are relevant."

4) The authors mention in the text some Zfh1+ and Eya+ cells in reference to Figure 5D-E, but Eya staining is not shown in the figure.

We corrected the manuscript accordingly: “We stained for markers of differentiation and found that most cells expressed Zfh1, Eya and no or very few expressed the stalk cell marker LamC (Fig. 6F-K).” Fig. 6F-G show Zfh1 immunostaining, while Fig. 6H-I show Eya immunostaining.

5) There is a reference to Figure S6K in the text but Figure S6 stops with the H panel. In addition, the figure legend goes up to panel L.

The legend was corrected accordingly.

Reviewer #5 (Remarks to the Author):

We thank the reviewer for their time and input!

Response to referees - NCOMMS-25-39031A

While we did not receive additional input from Reviewer #3, we would like to thank them for their time and consideration.

Reviewer #1 (Remarks to the Author):

This revised manuscript represents a substantial and thoughtful improvement over the previous version. The authors have clearly invested significant effort in addressing both conceptual and technical concerns, particularly by restructuring the Results, narrowing the focus to Hh and EGFR–Ras signaling, and adding extensive new experimental data.

The additional epistasis experiments and quantitative pathway reporter analyses considerably strengthen the claim that Hh and EGFR–Ras do not cross-regulate each other at the level of pathway activation. Similarly, the inclusion of pseudotime analyses, mapping confidence scores, staged phenotypic analyses, and quantitative rescue experiments markedly improves the rigor of the single-cell and genetic conclusions. The new data supporting an EMT-like phenotype downstream of Hh/Zfh1—combining transcriptional signatures with polarity defects and live imaging—are particularly compelling and elevate the mechanistic depth of the study.

We thank the reviewer for their judgement and their effort to revising this manuscript.

That said, a few conceptual vulnerabilities remain. Most notably, the manuscript continues to use the term “independent” in a way that may be interpreted more broadly than warranted by the data. While pathway activation appears independent, the strong transcriptional convergence and cooperative effects on proliferation and differentiation suggest functional convergence at downstream regulatory nodes. A slightly more cautious framing—especially in the Abstract and Discussion—would reduce the risk of overstatement.

We revised the title and text of the manuscript accordingly and now use the term “non-interacting” instead.

We would like to note, that we do not agree that our data provides direct evidence of transcriptional convergence. We identify distinct downstream transcriptional targets, including *zfh1* downstream of Hh signaling and *pnt* downstream of EGFR-Ras signaling. Further, while both pathways promote proliferation, in ectopic Hh signaling this is accompanied by blocked differentiation while it is uncoupled from the differentiation status downstream of EGFR-Ras signaling. This suggests distinct regulatory modes downstream of these pathways.

Likewise, although the evidence for tumor-like growth upon co-activation of Hh and EGFR–Ras is strong within the *Drosophila* ovary context, the use of the term “tumor” may still raise concerns for some readers. An explicit definition of what constitutes a tumor in this system, and a brief acknowledgment of which classical cancer criteria are and are not met, would improve clarity and pre-empt criticism.

We now use the word tumor more cautiously and instead refer to “tumor-like growth” or use “tumor model”. We also added a “Limitations of the Study” section to discuss which cancer hallmarks our data support and which remain uncertain:

“We find that follicle cells with co-overactivation of Hh and EGFR-Ras signaling fulfill multiple criteria of cancer. They are induced by overactivation of genes that are orthologous to well-known human oncogenes, loose normal tissue architecture, upregulate EMT, display metabolic deregulation, blocked differentiation and sustained proliferation. Further, they are ultimately lethal for the host organism. While co-overactivation will certainly prove useful as a tumor model, future work will need to address, which additional hallmarks of human cancer, including for example replicative immortality, genome instability and induction of inflammation⁸⁴, are met by this model.”

The identification of MB cells as the likely cell of origin for the overgrowth phenotype is a clear advance, enabled by the use of additional cell-type-specific Gal4 drivers. However, because this conclusion relies on driver specificity rather than lineage tracing, the inherent limitations of this approach should be stated explicitly in the manuscript itself.

We have added a respective statement in our new “Limitations of the study” section:

“Using cell-type specific Gal4-driver lines, we identified MB cells as the likely initiating cell population promoting tumor-like growth. This conclusion is limited by the specificity of the applied Gal4-driver tools.”

Finally, while the single-cell analyses are comprehensive and technically sound, the Results section remains analytically dense. Select consolidation of scRNA-seq figures—without reducing data availability—could further sharpen the central biological narrative.

While we understand that the sections of the manuscript addressing scRNA-seq data require some level of concentration for readers less familiar with such analyses, we feel that reducing explanation further would limit the informative value for readers interested in our application of scRNA-seq analysis tools. Each analysis tool provides slightly different insights into the biology of these signaling pathways. Further, we use reference mapping in an unconventional, yet highly constructive manner, showing that this algorithm is useful for scientists analysing transcriptional effects of mutants.

In summary, this is now a strong and carefully executed study that makes a meaningful contribution to our understanding of how parallel signaling pathways jointly constrain epithelial homeostasis. With modest adjustments to terminology, framing, and emphasis, the manuscript should be well positioned for acceptance.

We thank the reviewer for their support and are grateful for their contribution in improving this manuscript.

Reviewer #4 (Remarks to the Author):

The authors have drastically restructured the manuscript and restricted the scope of the study, which now focuses on Hh and EGFR signaling pathways, leading to an improved revised version that gained in clarity and interest. The authors have also made substantial efforts to add convincing new data and provide more detailed explanations of experimental procedures, therefore addressing all our concerns.

We thank the reviewer for their judgement.

Nevertheless, a couple of minor points should be addressed:

- We agree with Reviewer 2 that the title of the manuscript does not reflect correctly the results of the study. That Hh and EGFR pathways induce tumorigenesis when overactivated does not mean that their physiological role is to prevent tumorigenesis. It is quite the opposite that would be required to be able to draw that conclusion, i.e if a loss of pathway activity would lead to tumorigenesis. As it stands the title (but also some statements in the abstract and other parts of the manuscript) is misleading. The title is only true in the context of overactivation. In addition, it suggests that there is a mechanism in place to limit pathway activity, hence preventing tumorigenesis, but this is not investigated in this work. Same for the following section title: "Independent regulatory modes of proliferation prevents expansion of undifferentiated cells", this is true upon pathway overactivation.

We recognize this mistake and have changed the title and reworded manuscript accordingly. We now explain our view on the protective mechanism in more detail in the discussion:

“Co-overactivation of either combination of these non-interacting pathways results in a significant expansion of the FSC pools. At the same time the loss of the activity of one of these pathways does not halt epithelial cell production^{16,17}. Therefore, compartmentalization of proliferative induction to several non-interacting signaling pathways provides a protective mechanism during tissue homeostasis. It conserves proliferative activity in the event of pathway loss while also safeguarding cells from the effects of ectopic pathway activity that may be caused, for example, by mutations of proto-oncogenes or tumor suppressors.”

Along the same line, why the authors undertook a gain-of-function approach in the first place is not clear.

A major goal of this study was to understand how signals regulating proliferation and differentiation in the early FSC lineage interact. Previous work primarily used loss- or reduction-of-function approaches to establish the necessity of these pathways for FSC self-renewal, proliferation, and differentiation. Our analysis of pathway overactivation complements this by testing the sufficiency of EGFR–MAPK and Hh signaling and examining whether their spatial restriction is required for proper lineage behavior.

Although both pathways are required for normal proliferation, activation of either alone was insufficient to drive lethal tumor-like overgrowth. In contrast, simultaneous activation of both pathways in main body cells produced lethal tumor-like growth, revealing their cooperative potential. These findings help explain the spatial logic of pathway activity: both signals promote high proliferation in FSCs, whereas downstream reduction of EGFR–MAPK activity permits differentiation induced by Hh pathway activity.

We added in the manuscript:

“While loss- and reduction-of-function analyses of EGFR-Ras and Hh signaling established their necessity for follicle epithelial proliferation and differentiation, it is less well understood how spatial restriction of pathway activity affects follicle epithelial homeostasis. Hence, in order to identify cell-type specific effects of pathway overactivation, we performed single-cell RNA-sequencing...”

- Also in the section "Independent regulatory modes of proliferation prevents expansion of undifferentiated cells", the conclusion « co-overactivation of independent signaling pathways protects the tissue from aberrant expansion of undifferentiated cells and therefore represents a safety mechanism for epithelial homeostasis » is wrong.

We agree and have revised the respective section in the manuscript.

- What are stalk cells in EGFRCA and RasV12 since follicles are fused?

For most of the manuscript we identified stalk cells bioinformatically or by marker expression. Therefore, we assume that the reviewer is concerned about the identification of stalk areas for the measurement of pathway activity in Figure 1K-L. For EGFR^{CA} ovarioles stalk areas are usually recognizable as morphologically distinct areas of follicle cells that accumulate between germline cysts in early stages (Figure 1I). Identification of stalk areas in Ras^{V12} ovarioles are somewhat more difficult, yet we can detect cells expressing stalk cell markers at areas where cells are not directly associated with germline cells and hence selected such areas. We describe in the legend of Figure 1:

“K-L) Quantification of Ptc-cherry (K) and dpERK (L) activity. We normalized measurements from the second stalk or areas with stalk-like morphology to R1 areas for each quantified picture.”

- Panels C and Y from Figure 5 are very difficult to read with the color code used. Comparison of average expression in the different conditions with wild-type is not possible.

We now present the quoted plots with the same color scheme for all genotypes.

We again want to thank their reviewer for very productive and very thoughtful suggestions that have helped us to improve the manuscript.

Reviewer #5 (Remarks to the Author):

We thank the reviewer for their time and contribution.